# Proximal tubular dysfunction as a predictor of AKI in Hospitalized COVID-19 patients

**Amit Bari[1], Muhammad Rafiqul Alam[2], Sumona Islam** 🄸[3]*, **Muhammad Nazrul Islam[2], Md. Omar Faroque[2], Noureen Amin[2]**

1 Department of Nephrology, Kidney Foundation Hospital and Research Institute, Dhaka, Bangladesh,
2 Department of Nephrology, Bangabandhu Sheikh Mujib Medical University, Dhaka, Bangladesh,
3 Department of Gastroenterology, Delta Medical College and Hospital, Dhaka, Bangladesh

* sumona.islam.borsha@gmail.com

**Data Availability Statement:** Deidentified dataset has been made available in the biobank repository, with the public availability set at 2024-06-30. URL: https://www.ebi.ac.uk/biostudies/studies/S-BSST1269 Accession number: S-BSST1269

## Abstract

### Background

High concentration of Angiotensin converting enzyme receptors in the proximal tubules make kidneys an early target in COVID-19. Proximal tubular dysfunction (PTD) may act as an early predictor of acute kidney injury (AKI) and more severe disease.

### Methods

This prospective observational study was conducted in the COVID unit, Bangabandhu Sheikh Mujib Medical University. 87 COVID-19 patients without known kidney disease were screened for 6 markers of PTD on admission–hyperuricosuria, normoglycemic glycosuria, proteinuria, renal phosphate leak, sodium leak and potassium leak. Positivity of 2 of the first 4 markers was considered as PTD. 35 patients with PTD and 35 without PTD were followed up throughout their hospital stay.

### Results

52.9% had PTD on admission. The most prevalent markers were renal sodium leak (67%), followed by proteinuria (66.7%), hyperuricosuria (42.5%), potassium leak (32.2%), phosphate leak (28.7%) and normoglycemic glycosuria (20.7%). Mean age was 55.7 years. 32.9% patients developed AKI. PTD group had higher odds of developing AKI (odds ratio 17.5 for stage 1, 24.8 for stage 2 and 25.5 for stage 3; p<0.0001). The mean duration of hospital stay was 9 days higher in the PTD group (p<0.001). PTD group also had higher odds of transferring to ICU (OR = 9.4, p = 0.002), need for mechanical ventilation (OR = 10.1, p = 0.002) and death (OR = 10.3, p = 0.001). 32.6% had complete PTD recovery during follow-up.

### Conclusion

Proximal tubular dysfunction is highly prevalent in COVID-19 patients very early in the disease and may act as a predictor of AKI, ICU transfer, need for mechanical ventilation and death.

**Funding:** The author(s) received no specific funding for this work.

**Competing interests:** The authors have declared that no competing interests exist.

## Introduction

First identified in Wuhan, China in December 2019, after more than three years, the World Health Organization (WHO) announced COVID-19, the latest pandemic, caused by the novel coronavirus to no longer constitute a public health emergency of international concern (PHEIC) [1]. One silver lining to this devastation of unprecedented magnitude is the generation of large amount of data. As the dust settles, hopefully we can use them to better prepare us for the inevitable next pandemic.

Primarily known to be a respiratory pathogen, over time the coronavirus has shown to be able to infect almost every organ systems of the body [2]. Kidneys are one of the commonly affected organs in COVID-19. While the initial studies, mainly originating in China reported kidney involvement to be uncommon, subsequent research has shown much higher likelihood [3–9]. Kidney involvement mostly takes the form of proteinuria, acute kidney injury (AKI), AKI on top of chronic kidney disease (CKD) and electrolyte imbalance [10–16]. Apart from posing diagnostic and therapeutic challenges regarding COVID management, kidney involvement has also shown to have prognostic value. Patients with AKI went on to have more severe disease, needed more ICU admission and ventilator support, have longer hospital length of stay and increased mortality [13, 17–19].

The presence of high concentration of Angiotensin Converting Enzyme (ACE) receptor in the proximal kidney tubules together with ultrastructural evidence of virus like particles consistent with severe acute respiratory syndrome coronavirus 2 (SARS-CoV-2) points towards early involvement of proximal kidney tubules in COVID-19 [12, 20–25]. Since proximal renal tubules is the site of reabsorption of the majority of substances filtered by the glomerulus, its involvement gives rise to disproportionate excretion of these substances in the urine, mostly phosphate, uric acid, protein, typically low molecular weight proteins like β2 microglobulin, glucose, sodium, potassium, amino acid and bicarbonate [26, 27]. Thus, disproportionate excretion of these substances may act as markers of proximal tubular dysfunction (PTD) in COVID-19, giving rise to a picture similar to complete or partial Fanconi syndrome (20). Since these changes occur earlier in the disease process it may act as a predictor of AKI and subsequently more severe disease [20, 28].

## Methods

This was a prospective comparative study that was carried out from May 2021 to September 2021 in the COVID unit and intensive care unit (ICU), Bangabandhu Sheikh Mujib Medical University (BMMU), Bangladesh. Eighty-nine reverse transcriptase polymerase chain reaction (RT-PCR) positive COVID-19 patients aged 18 years or above, with no known chronic kidney disease (CKD) admitted in the BSMMU COVID unit (not needing ICU support on admission) during this period were initially recruited. Serum creatinine and 6 markers of PTD–hyperuricosuria, renal phosphate leak, proteinuria, normoglycemic glycosuria, renal sodium leak and renal potassium leak were checked at baseline, which was within 48 hours of admission.

Since studies looking at proximal tubular dysfunction were limited, properly validated diagnostic criteria for proximal tubular dysfunction in COVID-19 patients were not available. Patients with at least 2 of the following 4 abnormalities–inappropriate uricosuria, renal phosphate leak, normoglycemic glycosuria and proteinuria were considered to have proximal tubular dysfunction, a criterion adopted from Kormann et al. [20] (Operational definitions).

The first 35 patients with PTD and 35 patients without PTD were included for follow-up, which was done twice weekly for clinical progression and weekly for resolution of PTD throughout their hospital length of stay. Patients who had raised serum creatinine at baseline were considered to have undiagnosed CKD or community acquired AKI (CAAKI) and not

included in any of the analyses (Operational definitions). Patients who withdrew from the study, got shifted to a different hospital or got discharged against medical advice (AMA) were dropped out of the follow-up portion of the study. The drop-outs were addressed by dynamic sample recruitment to reach 35 samples in both the PTD and the non-PTD group. Outcomes were measured as development of acute kidney injury (AKI), initiation of renal replacement therapy (RRT), shift to ICU, complete or partial recovery of kidney function (including recovery from PTD) or death (Operational definitions).

The statistical analysis was done by statistical analysis software (SAS) studio version. The difference in means or percentages of different variables was calculated using either the chi-square test for nominal variables or the Unpaired student t-test for numerical variables. For the analysis of outcomes, groups were compared using univariate and multivariate linear and logistic regression models. P-values less than 0.05 were considered significant.

The research study described in this article adheres to ethical guidelines and principles, receiving approval from the Institutional Review Board (IRB) of Bangabandhu Sheikh Mujib Medical University with approval number 3567. Prior to their participation, all patients involved in the study provided informed written consent, ensuring their voluntary participation and protection of their rights and privacy.

## Results

**Fig 1** describes the initial sample recruitment, follow-up and drop-outs. The mean age of the patients was 55.7 years. Of them 72.9% were male. Diabetes was the most common comorbidity, found in about half of the patients, closely followed by hypertension (47.1%). Mean Charlson Comorbidity Index score was 3.7. The mean duration from symptom onset to hospital admission was 6 days. The mean duration from hospital admission to screening for kidney function and PTD was 1.4 days.

The patients who developed PTD were older and had higher educational status (p = 0.04, 0.002). Other demographic parameters, comorbidities and COVID-19 related symptoms on admission did not vary between the 2 groups.

**Fig 2** shows the frequency of proximal tubular dysfunction among the 87 Covid 19 patients with normal kidney function screened for the 6 markers of PTD on admission. Renal sodium leak seemed to be the most common PTD, present in 67% of patients. Hyperuricosuria, renal phosphate leak, proteinuria and normoglycemic glycosuria were the 4 defining markers of PTD. 52.9% were found to have at least 2 of these positive at admission. Complete recovery of PTD was seen in 32.6% patients during their hospital stay. Mean time to recovery was 13.9 days.

**Table 1** demonstrates the markers or predictors of disease severity, most commonly associated with COVID-19 infection. Most markers were significantly higher in the PTD group compared to the non PTD group. 67 out of the 70 patients had severe disease. Due to the very high percentage of severe cases, disease severity was not considered in the analyses.

Twenty three out of the 70 patients (32.9%) who were followed up, developed AKI during their hospital stay. 21 were from the PTD group and 2 from the non PTD group. 9 patients developed stage 1 AKI (12.9%). 5 developed stage 2 AKI (7.1%) and 9 patients developed stage 3 AKI (12.9%). **Fig 3** shows that among the 21 AKI patients who had PTD on admission, 47.6% had complete recovery of kidney function, 19.1% had partial recovery of kidney function and 33.3% died during their hospital stay. Among the 2 AKI patients who did not have PTD on admission, one had complete recovery of renal function during the hospital stay and one had no recovery of renal function. 6 out of the 23 (26.1%) patients with AKI required renal replacement therapy in the form of hemodialysis.

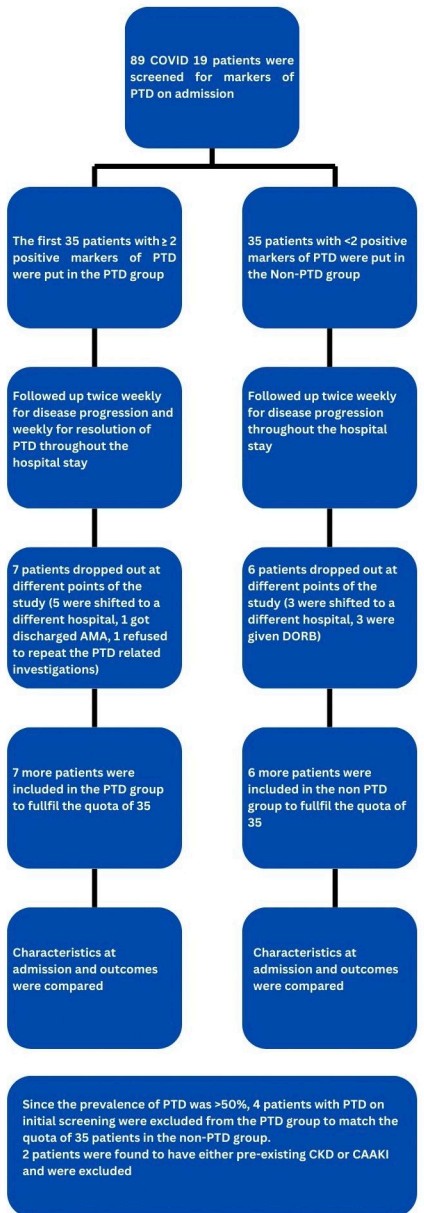

**Fig 1. Flowchart describing the study procedure (DORB = Discharge on risk bond, CAAKI = Community Acquired Acute Kidney Injury).**

The relationship between proximal tubular dysfunction and advancing stages of AKI is demonstrated through a logistic regression model in **Fig 4.** After a univariate model showed a positive correlation, a multivariate model was constructed, controlling for the other important predictors for hospital acquired AKI. The multivariate model showed that the patients with PTD on admission had 18-fold increased odds of developing stage I, 25-fold increased odds of developing stage II and 26-fold increased odds of developing stage III AKI compared to the non PTD group even after controlling for hypotension, ICU transfer and need for mechanical ventilation. The area under the curve (ROC) value was 90.3%, indicating the PTD to be a strong predictor of AKI.

**Fig 2. Proximal tubular dysfunction: Frequency and recovery** (Frequency = Measured among the total patients screened on admission (n = 87), Recovery = Measured upon follow-up of the patients with the specific dysfunction; 2_PTD = Presence of at least 2 markers of PTD (52.9%), HU = Hyperuricosuria (42.5%), NG = Normoglycemic glycosuria (20.7%), Prot = Proteinuria (66.7%), RKL = Renal potassium leak (32.2%), RNL = Renal sodium leak (67%), RPL = Renal phosphate leak (28.7%).

The univariate linear regression model exploring the relationship between proximal tubular dysfunction and hospital length of stay showed that patients with PTD on admission on an average had to stay in the hospital 9 days more than patients without PTD. However, after a

**Table 1. Markers/predictors of disease severity in the PTD and non PTD group following admission.**

| Variables[a,b] | Total (n = 70) | PTD (n = 35) | Non PTD (n = 35) | p-value |
|---|---|---|---|---|
| SpO2 | 86.1 (6.6) | 83.4 (7.6) | 88.8 (3.9) | 0.0004* |
| qSOFA | 1.0 (0.8) | 1.2 (0.9) | 0.8 (0.7) | 0.03* |
| Chest CT | 43.9 (18.7) | 50.9 (17.2) | 36.4 17.6) | 0.002* |
| CRP | 96.0 (72.7) | 124.7 (82.9) | 61.5 (36.3) | 0.0008* |
| S. Ferritin | 1873.6 (1776.7) | 2351.8 (1448.3) | 1353.9 | 0.051 |
| D-Dimer | 2.7 (2.8) | 2.7 (2.7) | 2.7 (3.0) | 0.94 |
| LDH | 318.16 (54.0) | 292.3 (30.9) | 374.2 (51.9) | 0.0005* |
| Procalcitonin | 0.6 (0.3) | 0.8 (0.4) | 0.6 (0.2) | 0.13 |
| Percent Lymphocyte Count | 13.4 (6.7) | 12.4 (6.4) | 14.5 (7.0) | 0.2 |

[a]Data is expressed as mean (SD)

[b]Unpaired T-test

*Significant

SpO2 = Oxygen saturation on room air measured on admission

qSOFA = Quick sequential organ failure score measured on admission. Rest of the investigations were done over the first 72 hours of admission

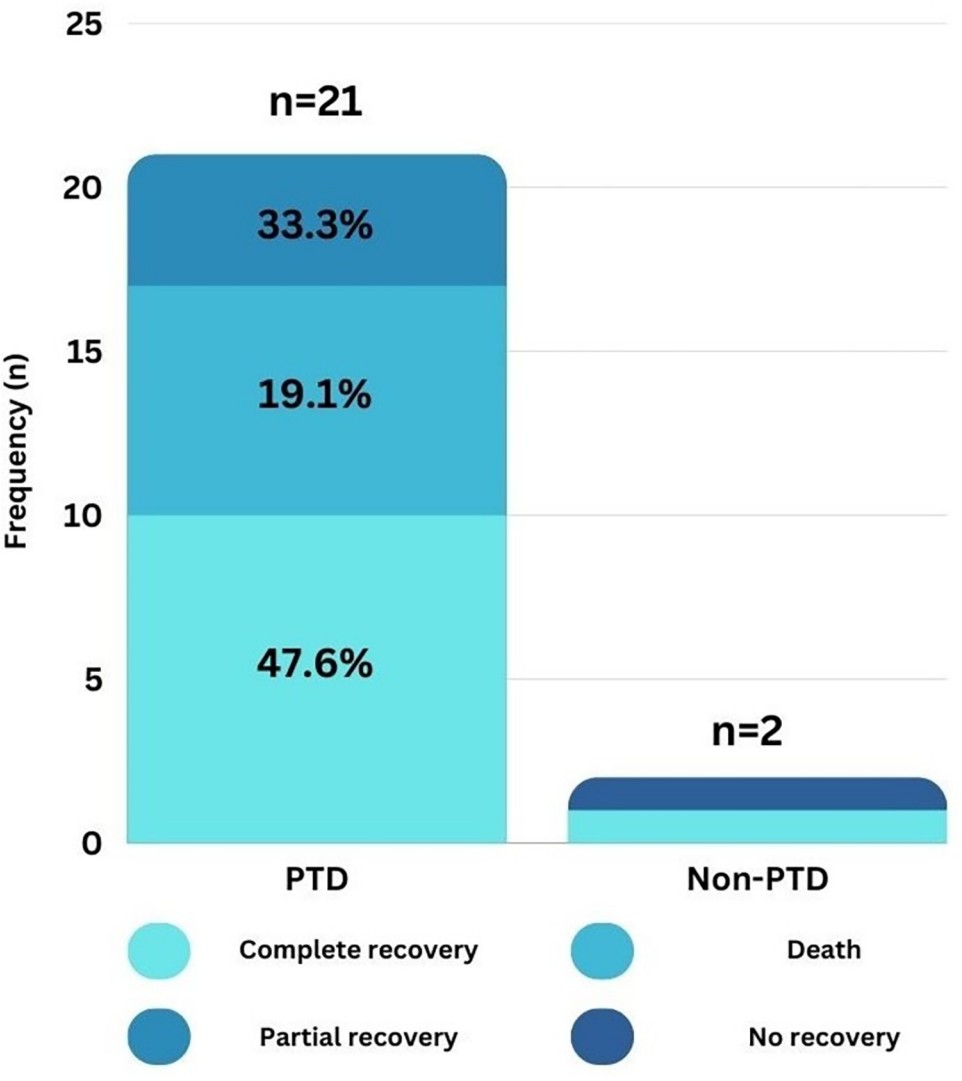

**Fig 3. AKI: Frequency and recovery separated according to the PTD on admission.**

multivariate model was constructed, controlling for development of AKI, transfer to ICU and need for mechanical ventilation the relationship did not persist, indicating that the increased LOS in the PTD group could be explained by these factors. The association is weak with an adjusted R-square of 0.15, indicating other potentially unexplored factors predicting the LOS.

Patients with PTD on admission also had higher odds of transferring to ICU, need for mechanical ventilation and death (OR = 9.4, p = 0.002; OR = 10.1, p = 0.002; OR = 10.3, p = 0.001) by univariate logistic regression analysis.

## Discussion

In this prospective comparative study, 87 hospitalized COVID-19 patients without preexisting CKD or CAAKI were screened for 6 markers of proximal tubular dysfunction–hyperurico-suria, renal phosphate leak, proteinuria, normoglycemic glycosuria, renal sodium leak and renal potassium leak on admission. The first 4 markers were used to develop the diagnostic

**Predicted Cumulative Probabilities for AKI_stage**
At ICU=0.243 Mech_vent=0.171 Hypotension=0.143

**Fig 4. Multivariate logistic regression model showing relationship between PTD and AKI stage after controlling for hypotension, ICU transfer and need for mechanical ventilation; Patients with PTD on admission have 18-fold increased risk of developing AKI stage 1, 25-fold increased risk of developing AKI stage 2 and 26-fold increased risk of developing AKI stage 3 compared with patients without PTD; Odds ratio = 17.5, 24.8 & 25.5 respectively, p < 0.0001 (0 = No PTD, 1 = PTD).**

criteria for PTD since similar diagnostic criteria has been used in previous studies [20]. Renal sodium and potassium leak were included as supplementary evidence.

At least 2 of the 4 defining markers of PTD was found to be positive in 52.9% of COVID-19 patients. This was lower compared to the 68% found in the previous study that developed and used this diagnostic criterion. Earlier screening, within a mean 1.4 days after admission compared to the mean 5.4 days in the prior study, thus allowing a shorter period to develop PTD can explain this discrepancy [20].

Among the PTD defining markers, proteinuria measured by 24-hour urinary total protein (UTP) was the most common, found in 66.7% of the patients. Proteinuria varied across studies from 20.3% to 85% [21, 26, 27]. Hyperuricosuria was found in 42.5% of the patients which ranged from 40% to 46% in previous studies [20]. We found renal phosphate leak in 28.7% patients. This varied between 19% and 39.6% across studies [20, 21]. Normoglycemic glycosuria was quite uncommon, seen in 20.7% patients. This was quite variable in prior studies, ranging from 0% to 28% [20, 21, 26]. We found that although glycosuria was quite common (48.6%), the high frequency of hyperglycemia (68.6%) lead to this lower frequency of normoglycemic glycosuria. It is possible that diabetes being a comorbidity in half the study subjects and widespread steroid usage led to this effect. We could not find studies that looked into renal sodium and phosphate leak in COVID-19. However, one study reported the prevalence of hyponatremia to be 25.8%, which was much lower than our study prevalence of 80% [26]. At 67%, we found renal sodium leak to be the most prevalent PTD. Renal potassium leak was found in 32.2% patients.

The widely used predictors of disease severity were also measured on admission. The level of CRP, LDH, SpO2, percentage of lung parenchymal involvement on HRCT and qSOFA score was significantly higher in the PTD group, which was in line with the previous studies [20, 21, 26]. No difference was found between D-dimer, Ferritin and percent Leukocyte count, which was inconsistent with prior studies. The early nature of our study can explain these findings.

Twenty three out of the 70 patients (32.9%) who were followed up, developed AKI during their hospital stay. 21 were from the PTD group and 2 from the non PTD group. Among the AKI patients who had PTD on admission, 47% had complete recovery of kidney function, 21% had partial recovery of kidney function and 32% died during their hospital stay. 6 out of the 22 (27.3%) patients with AKI required renal replacement therapy (RRT) in the form of hemodialysis. Due to the retrospective nature of the previous studies looking into COVID and PTD, timely follow-ups were often not possible. AKI prevalence ranged from 22% to 50% [20, 21].

Our study findings were already showing that PTD was quite common in COVID, quite early in the disease course and patients with PTD had much higher occurrence of AKI. To further explore this association, a multivariate logistic regression model was constructed, which showed that patients with PTD on admission had 18-fold increased odds of developing stage I, 25-fold increased odds of developing stage II and 26-fold increased odds of developing stage III AKI compared to the non PTD group even after controlling for hypotension, ICU transfer and need for mechanical ventilation. The area under the curve (ROC) value was 90.3%, indicating PTD to be a strong predictor of AKI. The previous studies that looked into this were all retrospective in nature. It was not possible to start with early onset of PTD on admission and follow them up to see if they develop AKI and subsequently more severe disease. But they did find 88% of patients with stage 2 and 3 AKI to have PTD, which is consistent with our findings [20].

Individual PTD markers were found to have prognostic value in COVID in previous studies. Hyperuricosuria and renal phosphate leak were found to be associated with the development of more severe disease, as evidenced by higher markers of disease severity, need for invasive mechanical ventilation, hospital length of stay and death [21]. But the overall impact of PTD on COVID prognosis was not reported. We found that patients with PTD on admission had higher odds of transferring to ICU, need for mechanical ventilation and death. A univariate linear regression model showed that patients with PTD on admission on an average had to stay in the hospital 9 days more than patients without PTD. However, after a multivariate model was constructed, controlling for development of AKI, transfer to ICU and need for mechanical ventilation the relationship did not persist, indicating that the increased LOS in the PTD group could be explained by these factors.

This study has several strengths. Most importantly, due to its prospective nature, it was possible to screen for markers of PTD very early after admission and follow the separate groups throughout the hospital stay to observe the difference in their characteristics on admission, clinical progression, development of AKI, need for RRT, need for ICU transfer, need for mechanical ventilation, length of stay and mortality, as opposed to the prior studies. We also included renal sodium and potassium leak, which were not included in previous studies, but proved to be quite prevalent and can be used in future studies as early markers of PTD. A structured in hospital follow up also gave us a better picture of the rate of resolution of PTD and AKI in COVID-19 patients.

The study is not without its fair share of limitations. In our study, aminoaciduria and urinary β2 microglobulin levels were not measured due to difficulty in sample processing, unavailability of proper diagnostic tools and high cost. Although TCO2 levels were reported via serum electrolyte assay, comment on metabolic acidosis was not made due to inability to perform and promptly send arterial blood gas samples in a general ward setting in the midst

the pandemic. Early biochemical markers of AKI like NGAL or KIM-1 were not included due to resource constraints. IL-6 level was not yet available and was not done. It was not possible to perform kidney biopsy to demonstrate ultrastructural evidence of proximal tubular involvement since most patients were receiving blood thinners. It was also not possible to do post-mortem analysis due to resource constraints. The participants were not followed up beyond their hospital stay to evaluate the comprehensive timeline of resolution of proximal tubular and kidney dysfunction due to time and resource constraints. The single centered design, small sample size and lack of less severe COVID-19 cases make it less generalizable. The absence of a non-COVID control group makes the role of COVID-19 less specific.

COVID-19 vaccines have been engineered, tested, mass produced and put forward for the public in record time [29]. But this was far from a success story. COVID-19 has left its marks. But it has been able to generate an abundance of data. Moving forward, the best outcome the health systems and public health personnel can expect is to use these data, build on them to reinforce our surveillance system against infectious diseases and extrapolate them towards fighting other infectious diseases in the future. If COVID-19 and other viral illnesses can be proven to be associated with PTD very early in the disease, which can be diagnosed by cheap and widely available investigations, and PTD can act as a predictor of disease severity, then it can be used to triage and allocate resources in future pandemics.

## Conclusion

This study revealed that proximal tubular dysfunction was a frequent finding in COVID-19. It developed very early in the course of the disease. Resolution seemed to vary widely among patients and based on individual marker. Frequency of PTD seemed to correlate with few, but not all of the other more commonly used markers or predictors of disease severity. Proximal tubular dysfunction is an independent predictor of AKI in COVID-19, which in turn is a predictor of disease severity, length of stay, morbidity and mortality. Future prospective studies should look into other markers of PTD in COVID-19 along with their role in other viral infection associated AKIs.

## Supporting information

**S1 Checklist. STROBE 2007 (v4) statement—Checklist of items that should be included in reports of *cohort studies*.**
(DOCX)

**S1 File. Operational definitions.**
(PDF)

**S1 Data.**
(XLSX)

## Acknowledgments

Abstract was presented in WCN23 as ePoster and visual abstract. Abstract ID is WCN 23–0213.

Bari, A., Alam, M.R., Islam, S., Islam, M.N., Faroque, M.O., and Amin, N. "Proximal Tubular Dysfunction as a Predictor of AKI in Hospitalized COVID-19 Patients." Kidney International Reports, vol. 8, no. 3, Mar. 2023, article S431, doi: 10.1016/j.ekir.2023.02.970. PMCID: PMC10025593. WCN23-0213.

## Author Contributions

**Conceptualization:** Amit Bari, Muhammad Rafiqul Alam, Sumona Islam, Muhammad Nazrul Islam, Md. Omar Faroque, Noureen Amin.

**Data curation:** Amit Bari, Noureen Amin.

**Formal analysis:** Amit Bari, Muhammad Nazrul Islam.

**Investigation:** Muhammad Rafiqul Alam, Sumona Islam.

**Methodology:** Amit Bari, Sumona Islam, Md. Omar Faroque, Noureen Amin.

**Project administration:** Amit Bari, Noureen Amin.

**Resources:** Amit Bari.

**Software:** Amit Bari.

**Supervision:** Muhammad Rafiqul Alam, Muhammad Nazrul Islam.

**Validation:** Amit Bari, Noureen Amin.

**Visualization:** Amit Bari.

**Writing – original draft:** Amit Bari, Sumona Islam.

**Writing – review & editing:** Amit Bari, Muhammad Rafiqul Alam, Sumona Islam, Muhammad Nazrul Islam, Md. Omar Faroque, Noureen Amin.

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
