## [Decision Letter · Decision Letter 0]

18 Oct 2023

PONE-D-23-28812Proximal Tubular Dysfunction as a Predictor of AKI in Hospitalized COVID-19 PatientsPLOS ONE

Dear Dr. Islam,

Thank you for submitting your manuscript to PLOS ONE. After careful consideration, we feel that it has merit but does not fully meet PLOS ONE’s publication criteria as it currently stands. Therefore, we invite you to submit a revised version of the manuscript that addresses the points raised during the review process.

We look forward to receiving your revised manuscript.

Kind regards,

Omid Dadras, MD, PhD

Academic Editor

PLOS ONE

5. Please amend your list of authors on the manuscript to ensure that each author is linked to an affiliation. Authors’ affiliations should reflect the institution where the work was done (if authors moved subsequently, you can also list the new affiliation stating “current affiliation:….” as necessary).

Reviewers' comments:

Reviewer's Responses to Questions

**Comments to the Author**

1. Is the manuscript technically sound, and do the data support the conclusions?

Reviewer #1: No

Reviewer #2: Yes

Reviewer #3: Partly

2. Has the statistical analysis been performed appropriately and rigorously? 

Reviewer #1: No

Reviewer #2: Yes

Reviewer #3: Yes

3. Have the authors made all data underlying the findings in their manuscript fully available?

Reviewer #1: No

Reviewer #2: Yes

Reviewer #3: Yes

4. Is the manuscript presented in an intelligible fashion and written in standard English?

Reviewer #1: Yes

Reviewer #2: Yes

Reviewer #3: Yes

5. Review Comments to the Author

Reviewer #1: The article explores proximal tubular dysfunction as a predictor of AKI in hospitalized COVID-19 patients. While the authors hope to contribute to scientific knowledge, there are several flaws in the methodology that limits its validity and reliability:

1. How were the patients recruited? What is the criteria for admission for COVID 19? You noted that the patients in both groups had "severe disease", how was "severe disease" defined? Where was the study conducted? In the methods the authors state "This was a prospective comparative study that was carried out from May 2021 to September 2021 in the COVID unit and intensive care unit (ICU), Bangabandhu Sheikh Mujib Medical University (BMMU), Bangladesh" Were patients recruited both from the COVID unit and the intensive care unit? Needs clarification.

2. Definitions are needed for 'proximal tubular dysfunction' is this synonymous with Fanconi's syndrome?

3. How did you assess PTD? what assays were used for measurements of the lab indices used? How did you define hyperuricosuria, renal phosphate leak, sodium leak etc.

4. How was Acute kidney injury defined, or "complete or partial recovery".

Figure 1 showing the flow chart of how patients were recruited showed 7 and 6 patients were added to the PTD and non-PTD group respectively, how and why were these patients chosen?

Results were not detailed enough. figures did not have legends.

Some references are not well written, ref 2,11,17,18,21,27 are some examples.

The limitations of the study were not mentioned by the authors.

I have also made some comments in the word text, which I will attach to this submission.

Reviewer #2: The authors have investigated the incidence and risk factors of bevazicumab related proteinuria.

Outcome - an event of proteinuria within 12 months of the index day

definition of ProtU >= 1+, >= 10 mg/dL spot urine or >= 120 mg/day in 24 hour collection samples.

anti-hypertensive treatment was considered

ProtU was registered by grade

In Univariate analysis patients with ProtU showed higher systolic blood pressure (p<0.001) and higher diastolic blood pressure.

ProtU was associated with higher creatinine (p<0.009) and with lower eGFr (p=0.0009)

Predictors of ProtU were: ovarian cancer, malignant glioma, higher nursing dependency score, lower eGFR, plant alkaloid based treatment.

In this study the ocurrence of ProtU was 2 fold higher than that reported in a previous meta-analysis, and it was not correlated with the initial dose of bevazicumab.

The authors consider to be necessary to evaluate the relationship between the cumulative dose of bevazicumab and proteinuria.

Is there any explanation for the higher occurrence of ProtU in this study?

Reviewer #3: In the manuscript Bari et al have performed a prospective study of COVID-19 patients for proximal tubular dysfunction (PTD) and acute kidney injury (AKI) allowing early screening and continuous monitoring of patients. The study assessed six markers of proximal tubular dysfunction, including newly considered markers like renal sodium and potassium leak. Furthermore, the study included a structured in-hospital follow-up, providing insights into the resolution of PTD and AKI.

Moreover, the study provides data on PTD, AKI, and their prognostic implications in COVID-19 patients. However, below are the comments for consideration

1-The lower study's sample size might limit the generalizability of its findings to a broader population.

2-The absence of a non-COVID control group makes it challenging to discern if the observed markers are specific to COVID-19 or common in other conditions.

3-While the study emphasizes the importance of data usage for future infectious disease surveillance, it doesn't provide specific recommendations or strategies for implementing these insights in public health systems.

6. PLOS authors have the option to publish the peer review history of their article (what does this mean?). If published, this will include your full peer review and any attached files.

Reviewer #1: **Yes: **Ngozi Virginia Aikpokpo

Reviewer #2: **Yes: **Teresa Adragao

Reviewer #3: No

---

## [Author Response · Author response to Decision Letter 0]

26 Dec 2023

Reviewer #1: The article explores proximal tubular dysfunction as a predictor of AKI in hospitalized COVID-19 patients. While the authors hope to contribute to scientific knowledge, there are several flaws in the methodology that limits its validity and reliability:

1. How were the patients recruited? What is the criteria for admission for COVID 19? You noted that the patients in both groups had "severe disease", how was "severe disease" defined? Where was the study conducted? In the methods the authors state "This was a prospective comparative study that was carried out from May 2021 to September 2021 in the COVID unit and intensive care unit (ICU), Bangabandhu Sheikh Mujib Medical University (BMMU), Bangladesh" Were patients recruited both from the COVID unit and the intensive care unit? Needs clarification.

The patients for this prospective study was recruited from the COVID unit, Bangabandhu Sheikh Mujib Medical University. This was an approximately 280 bedded makeshift COVID unit that operated throughout the pandemic. There was an ICU in the same building that was converted into a dedicated COVID ICU during this period, which ran independently from the COVID unit, but frequently had patients shifted from and to it. The admission criteria in the COVID unit varied widely throughout the pandemic, but it was generally patients with suspected or confirmed COVID-19 cases based on clinical, laboratory and imaging evidence, who did not meet the requirement for an ICU admission.

The recruitment in this particular study was however much stricter. We only recruited patients who were COVID-19 RT-PCR positive, aged 18 or above, had no known chronic kidney disease and consented to participate in the study within 48 hours of admission (This has been elaborated in the methods section for clarification). 

Since the electronic medical record (EMR) system is non-existent in Bangladesh, 2 of the 89 patients who we screened in order to recruit in the study was found to have renal impairment on their initial evaluation. They were thus labeled as having either pre-existing CKD or community acquired AKI (CAAKI) and excluded (This is mentioned in the flowchart describing the study procedure).

Since, the major outcome variables were ICU transfer, need for mechanical ventilation and death, we recruited patients from the COVID unit exclusively, not the ICU. During the in-hospital follow-up, if any patients were transferred to the ICU, we followed them up there until discharge, shift or death (This has been elaborated in the methods section for clarification).

The definition of “severe disease” was adopted from the institutional classification of COVID-19 cases, where oxygen saturation of <94% on room-air as measured by a hand-held pulse oximeter was considered as “severe disease” in order to simplify triage and admission using available. Since the COVID-19 cases vastly outnumbered the available beds, the hospitals mostly admitted severe cases only, which has led to almost exclusively recruiting severe cases (This has been added in the methods section for clarification). 

2. Definitions are needed for 'proximal tubular dysfunction' is this synonymous with Fanconi's syndrome?

Proximal renal tubules are involved with reabsorption of the majority of filtered solutes in the kidneys. A dysfunction generally suggests a failure of this capacity and subsequent loss of the solutes in urine. Fanconi syndrome generally describes a global dysfunction of the proximal tubules, which can be either inherited or acquired. 

Since data regarding the extent and magnitude of the involvement of the proximal tubules in COVID-19 cases were quite scarce and the evaluation of the full extent of proximal tubular function beyond the scope of this study, the term “proximal tubular dysfunction” (PTD) was preferred over “Fanconi syndrome” here, in order to take a more conservative approach. 

3. How did you assess PTD? what assays were used for measurements of the lab indices used? How did you define hyperuricosuria, renal phosphate leak, sodium leak etc.

We looked at 6 markers of proximal tubular injury – hyperuricosuria, renal phosphate leak, proteinuria, normoglycemic glycosuria, renal sodium leak and renal potassium leak. We could only find one published article that provided a case definition of proximal tubular dysfunction in COVID-19 at the inception of the study and we decided to use it. Following is the case definition and the operational definition of the 6 markers of dysfunction – 

Proximal tubular dysfunction: Presence of at least two of the four abnormalities – inappropriate uricosuria, renal phosphate leak, normoglycemic glycosuria and proteinuria was labeled as PTD. Renal sodium leak and renal potassium leak were included as possible markers of proximal tubular dysfunction, but not included in the definition of PTD.

Hyperuricosuria: Defined by serum uric acid (SUA) levels <220 micromol/L (3.7 mg/dL) in men and <184 micromol/L (3.1 mg/dL) in women, and a fractional excretion of urate {[(urine uric acid/SUA)/(urine creatinine/ serum creatinine)] × 100 g } > 10%.

Renal phosphate leak: Defined by a ratio of TmPi/GFR <0.77 mmol/L. This ratio was calculated in two steps as follows: (a) calculation of the renal tubular reabsorption of phosphate (RTP) with the following formula: RTP = 1 – [phosphate clearance (CPi)/creatinine clearance (Ccr)]  � 100 and (b) interpretation of RTP value: if RTP � 0.86: TmPi/GFR = RTP � plasma phosphate (Pp, mmol/L) and if RTP > 0.86: TmPi/GFR = � � Pp with � = 0.3 � RTP/[1 � (0.8 � RTP)]. 

Hypophosphatamia was defined as a value under the laboratory threshold of 0.78 mmol/L or 2.4 mg/dL.

Normoglycemic glycosuria: was defined by a glycosuria of + on a urine dipstick test or urine routine examination (equivalent to > 15 mg/dL) and glycemia of < 180 mg/dL or 10 mmol/L.

At least 1+ glycosuria as measured by dipstick in the absence of hyperglycemia (RBS < 180 mg/dl or 10 mmol/L).

Proteinuria: was defined by a high urinary protein creatinine ration (PCR): Proteinuria > 300 mg/g.

Renal sodium leak: Fractional excretion of sodium > 1 

Renal potassium leak: 

If normokalemia, fractional excretion of potassium > 16% 

If hypokalemia, fractional excretion of potassium > 9.5% 

If hyperkalemia, fractional excretion of potassium < 10%

(The complete definition and operational definitions of the markers have been included in the appendix for clarification)

4. How was Acute kidney injury defined, or "complete or partial recovery"?

The KDIGO definition of acute kidney injury was used in this study, which is an abrupt (within 48 hours) reduction in kidney function currently defined as an absolute increase in serum creatinine of more than or equal to 0.3 mg/dl (≥ 26.4 μmol/l) within 48 hours, a percentage increase in serum creatinine of more than or equal to 50% (1.5-fold from baseline) within 7 days, or a reduction in urine output (documented oliguria of less than 0.5 ml/kg per hour for more than six hours).

Complete recovery of renal function: Decrease in serum creatinine to normal/baseline, along with improvement in urine output during hospital stay. 

Partial recovery of renal function: Improvement in renal function as determined by increase in urine output & a decrease in serum creatinine but serum creatinine level still above normal/baseline at the time of discharge.

Baseline serum creatinine: Lowest serum creatinine recorded in the first 48 hours of hospitalization.

(These operational definitions have been included in the appendix for clarification)

4. Figure 1 showing the flow chart of how patients were recruited showed 7 and 6 patients were added to the PTD and non-PTD group respectively, how and why were these patients chosen?

As mentioned in the methods section, a dynamic recruitment strategy was used in this study. 7 patients from the PTD group and 6 from the non-PTD group dropped out at different points of the study due to shift to different hospitals, getting discharged against medical advice, refusing to repeat the investigations etc. Since following up patients in other institutes were beyond the scope of this study, we recruited more patients to match our quota of 35 patients in both groups to compare. We acknowledge that there is a potential of bias since we are not including these drop-outs in the final analysis, but since the number of drop-outs were almost similar in both groups and and their baseline characteristics were comparable, we believe that the chances are low. 

4. Results were not detailed enough. figures did not have legends.

Figure legends have been provided at the end of the manuscript. We are a bit confused. Should they be included in the image files?

4. Some references are not well written, ref 2,11,17,18,21,27 are some examples.

Few of the references mentioned have been updated.

4. The limitations of the study were not mentioned by the authors.

The limitations have been added in details at the end of discussion, which should have been done in the first place. We apologize for not doing that.

Reviewer #2: The authors have investigated the incidence and risk factors of bevazicumab related proteinuria.

Outcome - an event of proteinuria within 12 months of the index day

definition of ProtU >= 1+, >= 10 mg/dL spot urine or >= 120 mg/day in 24 hour collection samples.

anti-hypertensive treatment was considered

ProtU was registered by grade

In Univariate analysis patients with ProtU showed higher systolic blood pressure (p<0.001) and higher diastolic blood pressure.

ProtU was associated with higher creatinine (p<0.009) and with lower eGFr (p=0.0009)

Predictors of ProtU were: ovarian cancer, malignant glioma, higher nursing dependency score, lower eGFR, plant alkaloid based treatment.

In this study the ocurrence of ProtU was 2 fold higher than that reported in a previous meta-analysis, and it was not correlated with the initial dose of bevazicumab.

The authors consider to be necessary to evaluate the relationship between the cumulative dose of bevazicumab and proteinuria.

Is there any explanation for the higher occurrence of ProtU in this study?

This seems to be a review for a different study. There has probably been a mix-up, we are afraid. 

Reviewer #3: In the manuscript Bari et al have performed a prospective study of COVID-19 patients for proximal tubular dysfunction (PTD) and acute kidney injury (AKI) allowing early screening and continuous monitoring of patients. The study assessed six markers of proximal tubular dysfunction, including newly considered markers like renal sodium and potassium leak. Furthermore, the study included a structured in-hospital follow-up, providing insights into the resolution of PTD and AKI.

Moreover, the study provides data on PTD, AKI, and their prognostic implications in COVID-19 patients. However, below are the comments for consideration

1-The lower study's sample size might limit the generalizability of its findings to a broader population.

Yes. We agree. We have included this in the limitation section.

2-The absence of a non-COVID control group makes it challenging to discern if the observed markers are specific to COVID-19 or common in other conditions.

Yes. We agree. Considering the pandemic, limited resources and isolation barriers, it was not possible for the authors to arrange simultaneous data collection from the non-COVID unit. We have included this in the limitation section.

3-While the study emphasizes the importance of data usage for future infectious disease surveillance, it doesn't provide specific recommendations or strategies for implementing these insights in public health systems.

A section has been included at the end of discussion and conclusion to elaborate our recommendations.

---

## [Decision Letter · Decision Letter 1]

25 Jan 2024

Proximal Tubular Dysfunction as a Predictor of AKI in Hospitalized COVID-19 Patients

PONE-D-23-28812R1

Dear Dr. Sumona Islam

We’re pleased to inform you that your manuscript has been judged scientifically suitable for publication and will be formally accepted for publication once it meets all outstanding technical requirements.

Kind regards,

Omid Dadras, MD, PhD

Academic Editor

PLOS ONE

---

## [Editor Report · Acceptance letter]

23 Mar 2024

PONE-D-23-28812R1 

PLOS ONE

Dear Dr. Islam, 

I'm pleased to inform you that your manuscript has been deemed suitable for publication in PLOS ONE. Congratulations! Your manuscript is now being handed over to our production team.

Kind regards, 

on behalf of

Dr Omid Dadras 

Academic Editor

PLOS ONE